# Addressing Threats and Ecosystem Intactness to Enable Action for Extinct in the Wild Species

**Sarah E. Dalrymple [1,2,\*], Thomas Abeli [2,3], John G. Ewen [2,4], Tania C. Gilbert [2,5,6], Carolyn J. Hogg [2,7], Natasha A. Lloyd [2], Axel Moehrenschlager [2], Jon Paul Rodríguez [8,9] and Donal Smith [4]**

[1] School of Biological and Environmental Sciences, Liverpool John Moores University, Liverpool L3 3AF, UK
[2] IUCN Species Survival Commission Conservation Translocation Specialist Group, Calgary, AB T2N 1N4, Canada
[3] Department of Science, Roma Tre University, 00146 Roma, Italy
[4] Institute of Zoology, Zoological Society of London, London NW1 4RY, UK
[5] Marwell Wildlife, Hampshire SO21 1JH, UK
[6] School of Biological Sciences, University of Southampton, Southampton SO17 1BJ, UK
[7] Faculty of Science, School of Life and Environmental Sciences, The University of Sydney, Sydney, NSW 2006, Australia
[8] IUCN Species Survival Commission, Caracas 1060, Venezuela
[9] Instituto Venezolano de Investigaciones Científicas (IVIC), Miranda 1204, Venezuela
\* Correspondence: s.e.dalrymple@ljmu.ac.uk

**Abstract:** The species listed as Extinct in the Wild (EW) in the IUCN Red List of Threatened Species consist of 84 plants and animals that have been lost from their indigenous range. EW species are therefore restricted to ex situ conservation facilities and often have populations founded with few individuals. Our analysis demonstrates that 60% of EW species are associated with ecoregions that have very low proportions of intact habitat. Furthermore, threats such as invasive species, pollution, and climate change affect just over half of EW species and compound the obstacles facing their re-instatement to the wild. Despite these bleak assessments, there are various options for EW recovery. We present five scenarios that encapsulate the circumstances facing EW species and suggest potential conservation action for each of these situations. We illustrate these scenarios using case studies of EW species that demonstrate how the various options of ex situ management, reintroduction, and assisted colonisation to new habitat can be used to address the very exacting requirements of EW species. Our aim is to present a broad review of the obstacles facing the recovery of EW species whilst inspiring action to prevent the extinction of the most imperilled species on the planet.

**Keywords:** conservation translocation; ex situ conservation; habitat intactness; IUCN Red List

## 1. Introduction

Of the 153,388 species that have been assessed according to the IUCN Red List of Threatened Species (hereafter referred to as the IUCN Red List), 42,100 are classified as threatened, but only 84 species are currently listed as Extinct in the Wild (EW) [1]. The primary reason for categorising species as EW occurs when the remaining individuals exist only as ex situ populations in facilities such as zoos, aquaria, botanic gardens, and seedbanks, and sometimes in private collections. Alternatively, populations of EW species may exist in the wild in their indigenous range if these populations are subject to management, the intervention effectively constituting ex situ conditions. Examples include the reintroductions of Christmas Island blue-tailed skink *Cryptoblepharus egeriae* into predator-proof enclosures [2] and the interrupted brome that exists in its indigenous range, but only with concerted management efforts to maintain it in situ [3]. Finally, EW status is assigned when species exist as a "naturalized population (or populations) well outside the past range" [4]. The definition of 'well outside' is perhaps open to interpretation, but

examples include the Japanese fish, kunimasu *Oncorhynchus kawamurae*, found only outside its indigenous range after a commercially motivated introduction outlasted the native population [5]. Regardless of the exact situation by which species are categorised as EW, preventing their extinction relies on human intervention and, consequently, they are susceptible to the vagaries of changing societal values and competition for limited resources.

The precarious existence of EW species [6] is compounded by the potential for such species to be overlooked by statutory mechanisms that prioritise in situ conservation action when species are limited only to ex situ conservation facilities [7]. In addition, the extent to which EW species are protected by ex situ approaches varies considerably [8] and the outcomes of translocations into the wild are often poorly studied [9], albeit with some notable exceptions [10]. Addressing these knowledge gaps by building an evidence base for successful conservation measures will benefit current and future EW species. In turn, we will learn important lessons to more effectively protect species that are still being found in the wild.

There are numerous examples of species that have been reduced to captive populations, but have then undergone successful recovery. Examples such as the European bison *Bison bonasus* [11] and Californian condor *Gymnogyps californianus* [12] demonstrate what conservation can achieve when bold and, perhaps, unorthodox approaches are implemented to bring about species recovery. We should look to these cases as aspirational and use them to galvanise the conservation community to emulate their success. The following review aims to look critically at the current options available for the recovery of EW species and makes the case for considering ambitious programmes of management. We collate existing information on the threats faced by EW species in their indigenous habitat and provide a broad framework for action that is applicable to all EW species, and potentially to those Critically Endangered (CR) species that are currently on a trajectory to EW classification.

## 2. Outlook for EW Species

The unifying condition for the application of EW status is the absence of wild populations in the indigenous range and, consequently, EW species recovery relies on translocation from ex situ facilities to suitable habitats in the wild. Previous research has evaluated the status of ex situ populations [6], but for recovery in the wild to occur, there must be a suitable habitat that will support all stages of a species' life cycle with minimal influence from threats that could reduce the fitness and survival of the translocated individuals. We conducted a desk-based assessment of the threats associated with the current set of EW species using existing schemes to define the extent of habitat loss and fragmentation (referred to as 'intactness'), and present a breakdown of threats according to the IUCN Red List accounts for each EW species [13].

### 2.1. Extinct in the Wild Species

The IUCN Red List EW species comprises 44 plants and 40 animals. Of the plants, one is a fern, five taxa are cycads, 30 are dicotyledonous angiosperms representing 17 families, and eight are monocots representing Poaceae (grasses) and four other families. The EW vertebrate animal taxa are made up of amphibians (*n* = 2), reptiles (*n* = 2), birds (*n* = 5), fish (*n* = 11), and mammals (*n* = 2). The invertebrates consist of one taxon each from the insects and isopods, and 16 molluscs, specifically three freshwater snails from the genus Aylacostoma and thirteen terrestrial snails of the genus Partula.

Whilst the published list of EW species forms the basis of our analysis in this paper, it is important to clarify that some of the EW taxa have uncertain status and/or have been shown to have changed in their status since the publication of their IUCN Red List EW classification. All EW species currently listed in the IUCN Red List have been included, but species that are thought to be erroneously classified as EW until their accounts are next updated have been flagged in Table 1. A full rationale for any deviations from the Red List accounts has been published in Smith et al. [6].

**Table 1.** Extinct in the Wild species, taxonomic grouping, and the assessment of ecoregion intactness in the former indigenous range derived from Beyer et al.'s (2020) analysis: 'H' denotes a high proportion of intact habitat (green shades), 'M' denotes a moderate proportion of intact habitat (yellow-green to yellow-orange shades), and 'L' denotes a low proportion of intact habitat with change over time conveyed as degrading ('−'), stable ('±'), or increasing ('+') intactness (dark orange to red shades). Species denoted as 'NA' did not have georeferenced occurrence records in GBIF that corresponded to an ecoregion classified by Beyer et al. (2020).

| Tax. Group | Species | Common Name | Ecoregion Intactness | |
|---|---|---|---|---|
| PLANTS | *Abutilon pitcairnense* | Yellow fatu | H± | |
| | *Agave lurida* * | | L− | |
| | *Aloe silicicola* | | L± | |
| | *Alphonsea hortensis* | | L− | |
| | *Amomum sumatranum* | | M± | |
| | *Arachis rigonii* | Manicillo | H± | |
| | *Bromus bromoideus* | Brome des Ardennes | L− | |
| | *Bromus interruptus* | Interrupted brome | L− | |
| | *Brugmansia arborea* | Huanduj | L± | H± |
| | *Brugmansia aurea* | Huanduj | L− | M± |
| | *Brugmansia insignis* | Huanduj | L± | H− |
| | *Brugmansia sanguinea* | Huanduj | L− | H− |
| | *Brugmansia suaveolens* | | L− | M± |
| | *Brugmansia versicolor* | Huanduj | L− | H± |
| | *Brugmansia vulcanicola* | Guamuco | L− | |
| | *Camellia amplexicaulis* | | L− | |
| | *Corypha taliera* | Tali palm | L− | |
| | *Cyanea pinnatifida* | Haha; Sharktail cyanea | H− | |
| | *Cyanea superba* | Haha; Mt. Kaala cyanea; Superb cyanea | M± | |
| | *Cyrtandra waiolani* † | Fuzzyflower cyrtandra | M± | |
| | *Deppea splendens* | | L− | M− |
| | *Diplazium laffanianum* | Governor Laffan's Fern | L− | |
| | *Dombeya rodriguesiana* † | | L− | |
| | *Encephalartos brevifoliolatus* | Escarpment Cycad | L− | |
| | *Encephalartos heenanii* | Woolly cycad | L− | |
| | *Encephalartos nubimontanus* | Blue Cycad | L− | |
| | *Encephalartos relictus* | Swazi cycad | L± | |
| | *Encephalartos woodii* | Wood's cycad | L− | |
| | *Erythroxylum echinodendron* * | | L− | |
| | *Euphorbia mayurnathanii* * | Antique spurge | M± | |
| | *Franklinia alatamaha* | Franklin tree | L− | |
| | *Furcraea macdougallii* | Falso Maguey Grande | L− | |
| | *Kalanchoe fadeniorum* | | M± | |
| | *Kokia cookei* | Moloka'i treecotton | M± | |
| | *Lachanodes arborea* | She cabbage tree | NA | |
| | *Lysimachia minoricensis* | | L− | |
| | *Mangifera casturi* | Kalimantan mango | M± | H± |
| | *Mangifera rubropetala* | | L− | H± |
| | *Nymphaea thermarum* | | L+ | |
| | *Ochrosia brownii* | | H± | |

| Tax. Group | Species | Common Name | Ecoregion Intactness | |
|---|---|---|---|---|
| | *Rhododendron kanehirai* | | L− | |
| | *Senecio leucopeplus* | | L+ | |
| | *Sophora toromiro* | Toromiro | NA | |
| | *Trochetiopsis erythroxylon* | St Helena Redwood | L± | |
| **ISOPODS** | *Thermosphaeroma thermophilum* † | Socorro isopod; Socorro sowbug | M− | H± |
| **INSECTS** | *Leptogryllus deceptor* † | Oahu Deceptor Bush Cricket | L ± | |
| **MOLLUSCS** | *Aylacostoma chloroticum* † | | L± | M± |
| | *Aylacostoma guaraniticum* | | L+ | |
| | *Aylacostoma stigmaticum* | | No intactness assessment-EX | |
| | *Partula dentifera* * | Toothed partula | L− | |
| | *Partula faba* | Captain Cook's bean snail | No intactness assessment-EX | |
| | *Partula garrettii* | Garrett's tree snail | L− | |
| | *Partula hebe* | Tapairu tree snail | L− | |
| | *Partula mirabilis* | Navenave tree snail | L− | |
| | *Partula mooreana* | Eimeo tree snail | L− | |
| | *Partula navigatoria* | Raiatean ground partula snail | L- | |
| | *Partula nodosa* | Niho tree snail | L− | |
| | *Partula rosea* | Tarona tree snail | L− | |
| | *Partula suturalis* | Taamu tree snail | L− | |
| | *Partula tohiveana* | Tohiea tree snail | L− | |
| | *Partula tristis* * | Iareta tree snail | L− | |
| | *Partula varia* | Mauru tree snail | L− | |
| **AMPHIBIANS** | *Anaxyrus baxteri* | Wyoming toad | H− | |
| | *Nectophrynoides asperginis* | Kihansi spray toad | L− | M± |
| **FISH** | *Acipenser dabryanus* | Yangtze sturgeon | L− | |
| | *Allotoca goslinei* | Banded allotoca | M− | |
| | *Cyprinodon alvarezi* | Perrito de Potosi; Potosi pupfish | L+ | M+ |
| | *Cyprinodon longidorsalis* | La Palma pupfish | M+ | |
| | *Cyprinodon veronicae* | Charco Palma pupfish | M+ | |
| | *Notropis amecae* | Ameca shiner | L− | M- |
| | *Oncorhynchus kawamurae* | Kunimasu; black kokanee | L− | |
| | *Skiffia francesae* | Golden skiffia | L− | |
| | *Stenodus leucichthys* | Inconnu | L− | |
| | *Xiphophorus couchianus* | Monterrey platyfish | L+ | L± |
| | *Xiphophorus meyeri* | Marbled swordtail; Muzquiz platyfish | L± | |
| **REPTILES** | *Cryptoblepharus egeriae* | Blue-tailed skink | L− | |
| | *Lepidodactylus listeri* | Lister's gecko | L− | |
| **MAMMALS** | *Elaphurus davidianus* | Milu; Père David's deer | L− | |
| | *Oryx dammah* | Scimitar-horned oryx | L− | H± |
| **BIRDS** | *Corvus hawaiiensis* | 'Alalā; Hawaiian crow | L± | H− |
| | *Cyanopsitta spixii* | Spix's macaw | L± | |
| | *Mitu mitu* | Alagoas curassow | L− | |
| | *Todiramphus cinnamominus* | Sihek; Guam Kingfisher | L± | |
| | *Zenaida graysoni* | Socorro Dove | NA | |

All EW species currently listed in the IUCN Red List have been included, but species that are thought to be erroneously classified as EW are denoted with '†'. Species denoted with '*' are synonyms of species that are extant in the wild, as follows: *Agave lurida* now reclassified as *A. vera-cruz*, *Erythroxylum echinodendron* now reclassified as *E. minutifolium*, *Euphorbia mayurnathanii* now reclassified as *E. antiquorum*, *Partula dentifera* is now reclassified as *P. navigatoria*, and *P. tristis* is now reclassified as *P. garrettii*. The full rationale for presenting deviations to the published EW Red List is provided in Smith et al. [6].

### 2.2. Ecoregion Intactness

The obvious management option for EW species is to attempt a translocation to areas within the indigenous range, i.e., reintroduction. In addition to having suitable individuals for translocation and release/outplanting, reintroduction requires that the range is known and there is still suitable habitat available. Both of these requirements are typically challenging to meet for EW species. Normally, one might use previous records of the species' location to describe the historical range and many millions of such records are widely available online. However, EW species occurrence records are often limited in terms of describing former range due to the long timescales since some of these species were last seen in the wild (e.g., prior to 1950) [14] and the sometimes cryptic nature of the species. In order to assess the intactness of the available habitat consistently across all EW species whilst circumventing the problems caused by the patchy availability of occurrence records, we assumed that formerly occupied ecoregions [15] could be used to infer habitat intactness for EW species.

Using the Global Biodiversity Information Facility (https://www.gbif.org/, accessed on 2 August 2022), we downloaded all occurrence records for each of the EW species listed in the IUCN Red List (for R code, package citations and GBIF occurrence citations, please see the Supplementary Material documents S1 and S2). We created spatial point dataframes of each species' GBIF occurrence records and overlaid these points onto a global map of ecoregions [15] to identify the likely habitat and location. We used the R package {coordinateCleanR} [16] to remove duplicates, coordinate errors, outliers, and records erroneously associated with open sea and capital cities (often incorrectly logged when specimens are preserved in national collections). However, we did not choose to omit the records of preserved specimens and other non-observational records, instead opting to manually check the distribution maps and cross-reference these with IUCN Red List accounts and other authoritative descriptions of the species' distribution. For species with no GBIF records, we used the Red List accounts to manually assign the associated ecoregions.

Once the indigenous ecoregions had been identified, we collated estimates of habitat intactness using an approach developed by Beyer et al. (2020) [17] whereby habitat intactness is calculated from the habitat area, habitat quality as derived from the inverse of human pressure, and fragmentation of the habitat. Specifically, we utilised the nine-category system that arises from the factorial combination of three levels of available habitat (low, moderate, and high) for any given ecoregion, with three assessments of change (degrading, stable, and increasing) over the time period 1993–2009 [17]. Where a species was associated with more than one ecoregion and the intactness rating of those ecoregions differed, the intactness score for each ecoregion was recorded. The full dataset is available in the Supplementary Materials in file S3 with an accompanying Read Me file denoted as S4.

The EW species were identified from six biogeographical realms: Afrotropical (*n* = 10), Indo-Malayan (*n* = 9), Nearctic (*n* = 13), Neotropic (*n* = 22), Oceania (*n* = 22), and Palearctic (*n* = 8). Across those realms, 106 ecoregions were identified as being associated with the indigenous range of EW species. This number will be a conservative estimate of the number of ecoregions that have once supported EW species because the former distribution of some species is undescribed. Even for those EW species for which we do have records, it is very likely that the reported distribution is less extensive than the actual former range.

The species-specific assessments of ecoregion intactness are displayed in Table 1. Where species are associated with one category of ecoregion intactness (*n* = 64), this is displayed as a single code representing low (L), moderate (M), and high (H) proportions of intact habitat with its change of time indicated as increasing ecoregion intactness (+), stable intactness (±), and degrading intactness (-) and colour-coded accordingly. For species associated with more than one category of ecoregion intactness, the extremes are represented (*n* = 17). A number of species occupy ecoregions that were not included in Beyer et al.'s [18] analysis and are represented as 'NA'.

Fifty-one of the EW species (61%) are entirely associated with ecoregions containing a low proportion of intact habitat. Furthermore, the trend is towards further degradation, with 46% of all EW species associated with ecoregions that started with a low proportion of intact habitat and underwent further losses during the time period 1993–2009. Seven EW species are associated with ecoregions with a low proportion of intact habitat that was judged to be stable in terms of change over time, and a further three species are associated with ecoregions that showed improvement in ecoregion intactness albeit starting from a low baseline.

For the 17 EW species that are associated with multiple ecoregions of varying habitat intactness, the majority (*n* = 16) are associated with at least one ecoregion that has a high or moderate assessment of habitat intactness. Any attempt to translocate species to the wild would therefore have more options for exploring suitable release sites in areas where habitat is present with greater connectivity to facilitate future expansion. In summary, more than half of the EW species are faced with poor prospects in terms of identifying suitable habitat in the former native range, but for 25 species that are associated with one or more ecoregions that have moderate or high levels of intact habitat, their recovery in the wild is more feasible, at least from the point of view of habitat intactness.

### 2.3. EW Threats in Detail

Classifying species according to the likelihood of finding intact habitat is valuable for assessing the utility of reintroductions in progressing species recovery, but habitat quantity and quality are, of course, only part of the picture. Figure 1presents the threats identified in the Red List accounts of EW species [13] according to the number of times each threat is listed and presented by the timeframe over which Red List assessors reported the threats to be operating. A number of IUCN-recognised threats are implicit in the intactness analysis presented above (Section 2.2), e.g., urbanisation causes the loss of native vegetation and transportation corridors will result in fragmentation, both of which constitute a loss of 'intactness'. However, it is valuable to look at how threats that are unrelated to habitat loss are also frequent causes of the extirpation of EW species from the wild. For example, invasive species, genes and diseases, pollution, climate change, and geological events account for threats to 40 EW species. Because these threats operate independently to those that cause habitat fragmentation and loss, this finding further compounds any pessimism regarding the future of EW species; even where habitat availability is relatively good, a number of extensive threats might further complicate species recovery to the wild. Furthermore, the prevalence of threats that are listed as "past, unlikely to return" might be interpreted optimistically to mean that the threats would no longer cause a problem to EW recovery in the wild. Unfortunately, for many of these past threats, the damage has already been done and whilst threats such as agricultural conversion and natural system modifications are judged 'unlikely to return', it cannot be assumed that the habitat has been restored, nor that all adverse impacts have been resolved.

Invasive species, genes, and diseases is the threat category most frequently reported as affecting EW species being implicated in the extirpation of 35 species from the wild. The breakdown by timing of the threat indicates that in only seven cases have the problems associated with invasive organisms been controlled. However, the threat persists for 13 species for which invasive organisms are reported to be an ongoing problem, and for

another 15 species, the threat has been flagged as a potential future problem, either occurring anew, or likely to return after having been controlled or mitigated for in the past.

Biological resource use is next most frequently cited in IUCN Red List accounts of EW species and another example of a threat that does not directly contribute to the synthesis of ecoregion intactness developed by Beyer et al. [17] and presented above. Of the 27 species threatened by biological resource use (19 plants and eight animals), this threat is no longer thought to be currently operating for 13 taxa and is judged to have a low likelihood of becoming a problem again in the future. However, for two species, the Red List reports that the threat of biological resource use is in the past, but likely to return in the future. For the remaining 12 species, biological resource use is an ongoing problem that would need to be addressed prior to any attempt to translocate individuals back to the indigenous range.

Climate change and severe weather are reported as threatening only four of the EW species. This is unsurprising given that climate change has only recently featured prominently in IUCN Red List assessments, and many EW species have not been in their indigenous range for some time. As a future threat, it is likely that climate change may alter what available habitat exists and this should be incorporated into reassessments of threats and translocation feasibility assessments.

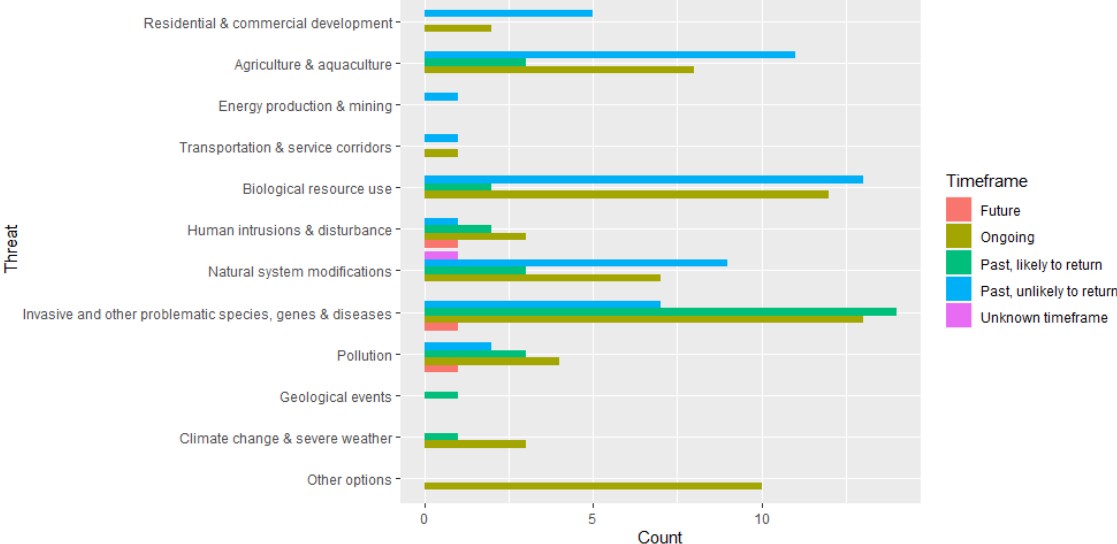

**Figure 1.** Frequency of threats cited in IUCN Red List assessments of EW species. Species are generally listed as being threatened by multiple processes, hence, the cumulative total of threat citations (*n* = 146) is higher than the number of species currently listed as EW. The timeframe is taken from the reported timing of threats in the IUCN Red List assessments. The timings of threats were not reported for ten EW species and were omitted from this dataset.

## 3. Options for EW Management

Our analysis of habitat intactness and the frequency of the cited threats demonstrates that there is potential for a wide range of obstacles to hinder species recovery, even within the relatively small group of 84 EW species. Whilst our assessment is novel—there have been no previous attempts to examine processes threatening EW species as a group—our evaluation consolidates the existing negative narratives around the potential for EW species recovery. To challenge this interpretation, the following section attempts to present scenarios that present alternatives for species management (Table 2). These scenarios avoid the assumption that all EW species require similar conservation action, and instead aim to highlight typical issues affecting this group of species so that solutions to specific problems can be presented. The scenarios presented in this paper are illustrated by case studies from the suite of EW species. We have selected species to represent a range of

taxonomic groups, locations, and contrasting situations. We do not attempt to be prescriptive for any one species, but rather try to challenge the perceptions of limited opportunity and inaction, and in doing so, dismantle the narrative that EW species are a lost cause.

**Table 2.** Scenarios encapsulating typical circumstances for EW-listed species with possible conservation actions that should be considered to facilitate EW species recovery.

| Scenario Summary | Explanation | Possible Management Options |
|---|---|---|
| A. Unknown cause(s) of loss in indigenous range. | The cause of extirpation from the wild is unknown and therefore cannot be targeted for management. | Maintain and increase the size of ex situ populations; consider trial reintroductions or assisted colonisation. |
| B. Management has minimised or eliminated threats. | The removal of the original cause of extirpation and/or minimisation of other threats have successfully reduced the impact of threats to a tolerable level that opens up the possibility for reintroduction. | Maintain ex situ populations; reintroductions; additional management to support population establishment post-translocation. |
| C. Substantive intervention required to minimise, mitigate, or eradicate threats. | Habitat exists, but is sub-optimal or intolerable without some sort of habitat restoration and/or threat management, and these interventions are judged to be exceptionally challenging. | Maintain ex situ populations; habitat restoration in situ; reintroduction once threats minimised and/or project commits to ongoing threat management; assisted colonisation. |
| D. Threats cannot be addressed within the indigenous range. | Habitat lost or rendered unsuitable by processes that could not be rectified by management and impossible to manage in situ e.g., climate change, habitat loss in its entirety, or presence of disease that is not survivable at population level. | Maintain ex situ populations; species-level interventions such as breeding or grafting for resilience to threats in indigenous range; assisted colonisation. |
| E. Causes of loss cannot be managed in indigenous range and assisted colonisation unsuitable. | Causes of loss are known, but cannot be managed and alternative options using conservation introductions are not available or judged unsuitable at the time of assessment due to (e.g.,) hybridisation risk, disease transmission, and/or detrimental impacts on the recipient ecosystem or species. | Maintain ex situ populations of the species; reassess suitability for release at regular time points in response to changes in the status of potential release sites within, and/or outside of, the species' indigenous range. |

## 3.1. Scenario A. Unknown Cause(s) of Loss in Indigenous Range

When the cause of extirpation from the wild is unknown or the species was not recorded in the wild so the cause of extirpation cannot be determined, it becomes extremely difficult to define targets for in situ management. The species must be maintained ex situ ensuring that populations are managed to improve genetic diversity and avoid problems such as hybridisation, inbreeding, and genetic drift. Releases/outplanting might be considered using robustly designed trials in a range of habitats thought to be suitable, which may be inside or outside the indigenous range. This may incur risks for the focal species and the recipient ecosystems, which should be assessed on a case-by-case basis and comprehensive monitoring programmes implemented according to IUCN guidance [19].

### 3.1.1. *Alphonsea hortensis*

This tree is endemic to Sri Lanka where it was known from lowland forest, but has not been recorded in the wild since 1969 and is listed at only one locality according to GBIF records [20]. The ecoregions with which it is associated are Sri Lankan lowland rain forests and dry-zone dry evergreen forests, both of which have a low proportion of intact habitat and are highly fragmented (see Table 1). The ex situ population is currently held in Peradeniya Royal Botanic Garden as a living collection, due to the seeds being recalcitrant (which prevents easy storage in seed banks). The historical major threats to the species are not known and, therefore, this species is a good example of scenario A, whereby threat eradication or mitigation is not possible and maintaining the species in ex situ facilities is currently the main focus of conservation action. In the longer term, outplanting trials to identify potential habitat requirements may offer a solution to re-establishing the species in the wild, depending on risk assessments.

### 3.1.2. *Camellia amplexicaulis*

Although this Vietnamese endemic shrub is frequently cultivated in its native country and beyond, there is no information as to why the species was lost from its only known wild locality in Tam Dao National Park, north Vietnam [21]. It is associated with the South China–Vietnam subtropical evergreen forests, which are classified as having a low proportion of intact vegetation and are undergoing further degradation [17]. The species exists in 14 formal ex situ collections worldwide [22]**,** but the horticultural interest in the plant means that many planted specimens outside of conservation collections may be cultivars unsuitable for planting in wild conditions. In some cases, living specimens thought to be examples of *C. amplexicaulis* have actually been described as new species [23]**,** highlighting the potential for taxonomic confusion in the ex situ collections that would need to be clarified prior to translocation.

We have selected this species as a case study for exemplifying scenario A because of the contrast with *Alphonsea hortensis*. Both species were extirpated from the wild due to unknown causes, but whilst *A. hortensis* presents problems for ex situ conservation due to difficulties in storing and propagating the plant, *C. amplexicaulis* is so common in cultivation that it might be difficult to identify suitable founder individuals. Possible solutions to this include propagation trials and experimental planting into the indigenous range to discern whether the species is still adapted to its nominal indigenous range.

### 3.2. Scenario B. Management Has Minimised or Eliminated Threats

The removal of the original cause of extirpation and/or minimisation of other threats are the key requirements for reintroduction and in this scenario will have occurred to reduce the impact of threats to a tolerable level, i.e., where the threat does not prevent population establishment or maintenance. Possible management options include the maintenance of ex situ populations to facilitate ongoing translocations over the long term and maintain genetic diversity, and reintroductions may be undertaken where a habitat is adequate to support viable populations. The species may also need additional management to support population establishment post-translocation and possibly into the longer term.

### 3.2.1. Milu (or Père David's Deer) *Elaphurus davidianus*

Milu is an example of a species having undergone reintroduction in partnership with management to overcome habitat degradation, and as such, constitutes a good case study for scenario B. If the reintroduction programme continues to be successful, the species is likely to lose its EW status in the next IUCN Red List reassessment. The species became extinct in the wild due to habitat loss and hunting, both of which have been addressed by releases of founders into reserve areas that can be monitored and managed appropriately [24]. It is associated with various flooded saline meadow and forest habitats in China, all of which are fragmented and limited in extent, but measures to extend the protected habitat and avoid concentrated grazing have seen the translocated populations increase in over 50 parks and reserves [25]. Reports of populations reaching the carrying capacity within protected areas were raised by Zhigang in 2013 [24] and were subsequently managed in some sites with supplementary feeding, although this has not prevented habitat degradation [25]. In several sites, milu have now dispersed from the original release areas and the reintroduction attempt has been so successful in promoting population growth that measures such as farmer's compensation, removal of animals, and even culling have been discussed [25].

### 3.2.2. Spix's Macaw *Cyanopsitta spixii*

Spix's macaw has recently been reintroduced to Brazil following measures to stem the key threats causing its extinction in the wild, and therefore constitutes another species that exemplifies scenario B. The species has been classed as EW since the last male individual in the wild disappeared in 2000, but substantial effort has been made to create an ex situ conservation breeding population. Although this effort is necessary for EW species, in this case, the attempt to create an ex situ population is controversial, having been built from private, and frequently illegal, collections. Those coordinating the reintroduction argue that since the threats from the pet trade and hunting have been made illegal, and the species has been adopted by local communities as a kind of mascot, the macaws now have a greater chance of escaping poaching in their wild habitat [26].

An additional and significant threat is habitat loss and attempts to address habitat loss in the Brazilian dry Caatinga forest are limited. However, Beyer et al.'s [17] intactness assessment indicates that the low proportion of native vegetation is at least stable over the time period 1993–2009 (Table 1). Work has been underway to conserve habitat in areas suitable for reintroduction, including managing the population of feral goats [27]. Novel release approaches using closely related species as 'mentors' for wild-living macaws are hoped to overcome the demographic and behavioural problems associated with the relatively small numbers of birds for release [26]. Individuals of Spix's macaw were released to their former habitat in June 2022 [28].

### 3.3. Scenario C. Substantive Intervention Required to Minimise, Mitigate, or Eradicate Threats

This scenario encompasses cases where habitat exists, but is sub-optimal without some sort of habitat restoration and/or threat management, and these interventions might be exceptionally challenging. As with other scenarios, ex situ populations need to be maintained, but in this case, the captive populations need to be kept in a condition that enables future translocations whilst restoration occurs in situ. Assisted colonisation to a new habitat might be considered if the indigenous range cannot be made suitable quickly enough. Alternatively, management may need to be ongoing to suppress the impacts of threats. As in scenario B, the species may also need additional management to support population establishment post-translocation and possibly into the longer term.

### 3.3.1. Scimitar-Horned Oryx *Oryx dammah*

This high-profile antelope from Saharan Africa might have been a good candidate for illustrating scenario B were it not for emerging threats that have been raised in the most recent IUCN Red List assessment [29]. The threats that caused the original species' extinction from the wild have been controlled to enable the release of substantial numbers of oryx to fenced protected areas in Tunisia [30] and Morocco, and a staggered release of >225 individuals to the unfenced Ouadi Rimé–Ouadi Achim Faunal Reserve (OROA) in Chad [31,32]. However, evidence of illegal trafficking and security risks in the region of the OROA reserve in Chad, means that efforts to conserve the species are exceptionally challenging. Although the population is meeting demographic indicators for success with the third generation born in the wild since the first release in 2016, and the most recent IUCN Red List account predicts downlisting in the next cycle of assessments, the prospects for long-term population growth in truly wild conditions are dependent on socio-political conditions in the region, which remain volatile.

### 3.3.2. Superb Cyanea *Cyanea superba*

As a Hawai'ian endemic plant, *Cyanea superba* is subject to many threats that are degrading the tropical forest ecoregions that form its native habitat with invasive alien plants, predation by feral pigs, rats and slugs, and wildfires started by a nearby military firing range acting together to cause direct mortality and habitat loss [33]. We have chosen this species as a further example of scenario C because these threats are deemed to be

"manageable albeit with enormous effort" [33]. Outplanting has been undertaken with mixed success due to ongoing herbivory by pigs, rats, and introduced slugs [34]. However, multiple controls for the various herbivores have enabled some individuals to survive to maturity.

### 3.4. Scenario D. Threats Cannot Be Addressed within the Indigenous Range

Scenario D describes a situation whereby the habitat has been lost or has been rendered unsuitable by processes that could not be rectified by management, e.g., climate change, habitat loss in its entirety, or the presence of a disease that is not survivable at the population level. As before, the maintenance of ex situ populations is crucial to facilitating ongoing translocations over the long term. Where habitat management or even species-level interventions such as breeding or grafting for resilience to threats are not an option, restoration to the wild will require the identification or creation of habitats outside of the species' indigenous range to enable assisted colonisation. Full risk assessment should be undertaken according to IUCN guidance on conservation translocations [19].

#### 3.4.1. Sihek *Todiramphus cinnamominus*

Sihek, a kingfisher endemic to the island of Guam, has been chosen to highlight scenario D because removing the cause of extinction, an invasive predatory snake, is currently extremely challenging, if not impossible, and this is compounded by the status of the ex situ population. In the time since the last wild individuals were brought into captivity, the ex situ population has reached capacity and is predicted to decline toward extinction without further intervention [35]. Identifying suitable habitat for sihek is therefore an urgent priority and needs to happen in a shorter timescale than is feasible for brown tree snake removal to be realistically be achieved. This has prompted alternative options to be sought, including proposed translocation to Palmyra Atoll, an island in the mid-Pacific that is free of snakes, with the aim of creating a wild-living population that can form the basis of reintroductions once the brown tree snake can be controlled on Guam.

#### 3.4.2. Christmas Island Blue-Tailed Skink *Cryptoblepharus egeriae*

The plight of the Christmas Island blue-tailed skink is very similar to that of sihek, and therefore has been selected as a second example of a species whose situation is typified by scenario D. The species underwent a very steep decline in numbers from being relatively common (albeit in a restricted range of Christmas Island) to the last individual being recorded in the wild in August 2010 [36]. The key cause of extirpation from the wild was predation by the Wolf Snake after its introduction in 1982, but other exotic predators, including yellow crazy ants, feral cats, rats, and an invasive centipede [2], are thought to have predated on the skink over longer time periods. In 2019, 300 skinks were released as an assisted colonisation to Pulu Ban, one of the Cocos (Keeling) Islands due west across the Indian Ocean from Christmas Island [37]. The island has been described as a refuge that will allow the species to establish a wild-living species away from invasive predators.

### 3.5. Scenario E. Causes of Loss Cannot Be Managed in the Indigenous Range and Conservation Introductions Are Unsuitable

When the causes of loss are known, but cannot be managed in situ, thereby excluding reintroduction, and alternative options using assisted colonisation are judged unsuitable at the time of assessment, the main conservation option is to maintain ex situ populations of the EW species to keep options open for future conservation intervention. Assisted colonisation might be unsuitable for a range of reasons including the risk of hybridisation, disease transmission, and/or detrimental impacts on the recipient ecosystem or species. It is recommended that suitability for release is reassessed at regular time points appropriate to the species and in response to changes in the status of potential release sites within and/or outside of the species' indigenous range.

3.5.1 Banded allotoca *Allotoca goslinei*

The banded allotoca, a fish known only from the Ameca River catchment, Mexico [38]**,** is thought to have been extirpated by a non-native and invasive fish (*Xiphophorus helleri*), probably through competition for similar prey and the predation of the allotoca larvae, although water pollution may have also caused declines prior to the introduction [38]. The difficulties of removing the invasive fish are too great for reintroduction to be a feasible option and the existence of a number of congenerics and subsequent risk of hybridisation prevents the assisted colonisation to nearby waterbodies. Consequently, this species must, for now, be restricted to ex situ conservation efforts including the Allotoca–Mesa Central breeding programme coordinated by the Austrian Association of Aquarists and a similar ex situ breeding project run by the Laboratorio de Biología Acuatica in Michoacan University, Morelia, Mexico [39].

*3.6. Adaptive Management*

The five scenarios presented in Table 2 can be set within the context of an adaptive management cycle developed by the IUCN SSC Conservation Translocation Specialist Group (Figure 2). The scenarios might inform project objectives and options for management, but as with any conservation intervention, good practice would also be informed by legislation, ecological knowledge of the species and recipient ecosystem, cultural understanding, and socioeconomic values. For example, the values and priorities of indigenous people and/or local communities should be incorporated into decisions and the tolerance to risks arising from various actions must be explored in consultation with all affected parties. These scenarios need not be mutually exclusive and indeed, the situations might be applied at different times to the same species or differentially applied across the species' former range, should this be large enough to encompass different ecoregions and threat exposure.

Figure 2 encompasses the next phases of modelling and the prediction of potential outcomes, which is followed by a process of addressing trade-offs between competing drivers and values. After deciding on a finalised plan and implementing the selected management interventions, a comprehensive monitoring programme should be implemented to describe the impact of any interventions and the impact of the focal species itself, as recommended by the IUCN Species Survival Commission [19]. If monitoring is limited to assessing the status of the focal species, important and potentially negative consequences of interventions might be missed. As with any adaptive management cycle, such as the Species Conservation Cycle followed by the IUCN Species Survival Commission [40], the monitoring data are used to reassess how well management objectives are being met and help guide what, if any, changes in management are required as critical knowledge is gained [41].

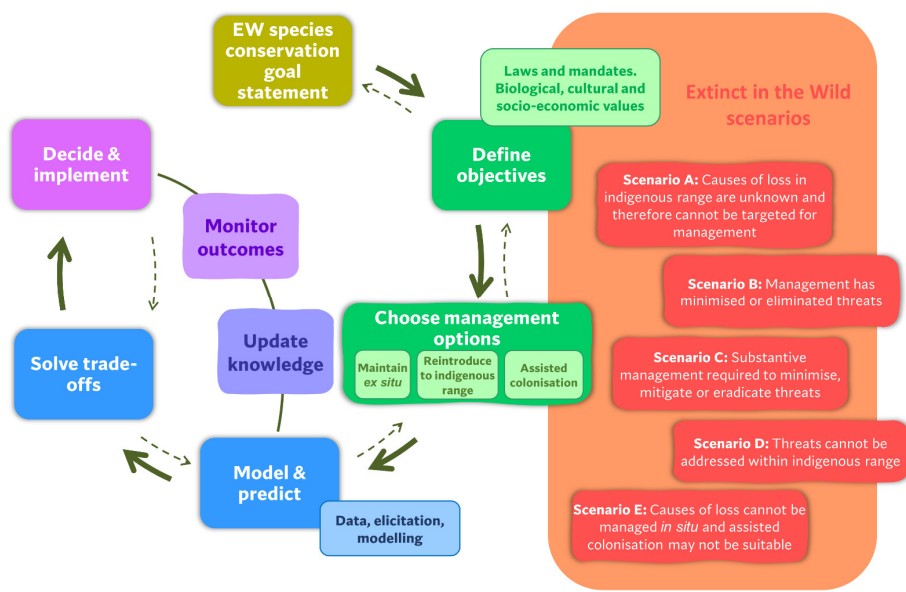

**Figure 2.** IUCN Conservation Translocation Specialist Group's structured decision-making cycle adapted to incorporate scenarios that exemplify the situation of Extinct in the Wild species.

## 4. Next Steps

Using our analysis of the habitat intactness of EW species and the approach developed by Beyer et al. [17], it is possible to identify species with a high proportion of intact habitat within their indigenous ecoregions (Figure 3). However, many of these 'intact' ecoregions are generally small in extent, being limited to regions of oceanic islands such as Pitcairn Island or Oahu, Hawai'ian Islands. Alternatively, species such as the Wyoming toad might have very specialist habitats within a much more extensive ecoregion such as the Wyoming Basin shrub steppe, which encompasses 130,000 km². Despite the expansive ecoregion, and an intactness score that indicates a high proportion of natural vegetation, the toad is effectively restricted to the floodplain ponds that it requires for egg-laying and tadpole development. For species that can be linked to extensive or multiple ecoregions, field surveys should be repeated to rule out the possibility that the species is still extant in the wild.

For those species that are associated with ecoregion intactness categories indicating a moderate proportion of natural vegetation, recovery prospects need to focus on protecting remaining habitat, restoring and creating habitat where possible. Even species that have very degraded ecoregions, with low levels of intact habitat, may be subject to translocation to small areas of suitable habitat when the focal species does not require extensive habitat (e.g., *Partula* snails). We recommend that opportunities for the recovery of EW species to the wild are thoroughly explored in every case, working within the constraints of site-based conservation, rather than using perceived limitations in habitat availability as a barrier to future action species' survival.

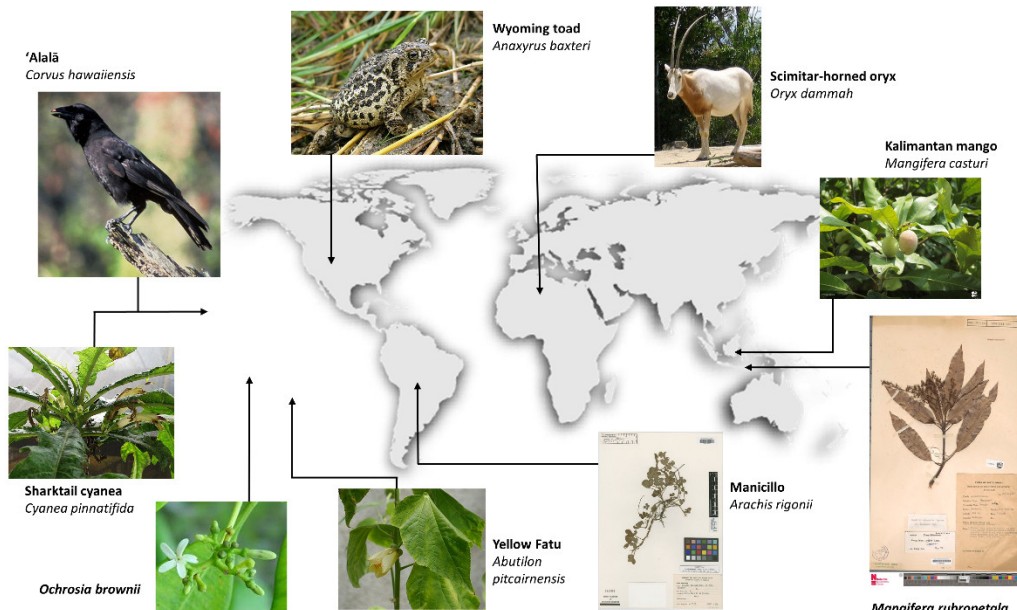

**Figure 3.** Extinct in the Wild species whose indigenous range is associated with ecoregions categorized as having a high proportion of intact habitat. Photo credits: *Abutilon pitcairnensis*: Salix, CC BY-SA 4.0 https://creativecommons.org/licenses/by-sa/4.0 accessed on 14 December 2022, via Wikimedia Commons; *Ochrosia brownii*: David H. Lorence, Jean-François Butaud, CC BY 4.0, via Wikimedia Commons; *Cyanea pinnatifida*: David Eickhoff from Pearl City, Hawaii, USA, CC BY 2.0 <https://creativecommons.org/licenses/by/2.0>, accessed on 14 December 2022 via Wikimedia Commons; Alala: U.S. Fish and Wildlife Service, public domain, accessed on 14 December 2022 via Wikimedia Commons; Wyoming toad: USFWS Mountain-Prairie, credit Sara Armstrong, CC BY 2.0 <https://creativecommons.org/licenses/by/2.0>, accessed on 14 December 2022 via Wikimedia Commons; Oryx dammah: Albinfo, CC BY-SA 3.0 <http://creativecommons.org/licenses/by-sa/3.0/>, accessed on 14 December 2022 via Wikimedia Commons; Kalimantan mango: https://toptropicals.com/catalog/uid/mangifera_casturi.htm; Mangifera rubropetala: Mangifera rubropetala Kosterm collected in Malaysia by Naturalis Biodiversity Center (licensed under http://creativecommons.org/publicdomain/zero/1.0/ accessed on 14 December 2022); Mancillo: Field Museum of Natural History—CC BY-NC 4.0, accessed on 14 December 2022.

## 5. Conclusions

Our review focuses on the conservation status and potential actions for EW species, but lessons can also be learned from this group for other threatened species. For example, for species that are declining, but still extant in the wild, the lack of planning in 'rescuing' species from the wild will cause problems for future management, whether this is because the collections preceded any conservation intention, or the rescue constituted the absolute last chance to save a species. Ex situ populations cannot easily be revitalised when captive individuals are from an unknown lineage and there are no remaining wild individuals to bolster dwindling genetic diversity. Subsequent releases will carry more uncertainties regarding the viability of translocated individuals, and as a result, may need more support after the individual plants and animals have been reinstated in the wild habitat. Alternatively, individuals may need to be released or outplanted under experimental conditions that explore ecological tolerances, but this also comes with drawbacks including the higher chance of mortality in conditions that turn out to be sub-optimal, and more demanding project management to ensure that trials are run rigorously. Unfortunately, we can safely assume that many more species face extinction from the wild and the dependence on ex situ populations will continue to grow; groups responsible for the conservation of extant in the wild species should look to the EW species to understand how high the stakes are in anticipating future declines.

Our analysis also highlights the need for greater exploration of conservation introductions, and specifically assisted colonisation, certainly for EW species, but perhaps also to maintain the viability of species that are extant in the wild [42]. Although such translocations are associated with greater uncertainty and, consequently, a higher risk of negative impacts, there is also risk in inaction of losing more species to the current extinction crisis. In today's volatile world, there is no such thing as a 'no risk' option when working with threatened species. Dismissing conservation introductions as a possible method for threatened species recovery risks consigning EW species to extinction. The fact that a similar number of species have become extinct as have achieved recovery in the wild since 1950 [6] should act as a stark warning that we cannot be complacent when species are assigned EW status. EW species provide some of the greatest challenges to conservationists, but conversely, may inspire our most ambitious interventions. The call to action is urgent and we encourage the conservation community to continue its efforts to save these imperilled species.

**Supplementary Materials:** The following supporting information can be downloaded at: www.mdpi.com/article/10.3390/d15020268/s1, GBIF citations for all occurrences used (S1), analytical script in coding language 'R' (S2), full dataset of all EW species and corresponding data (S3), 'ReadMe' file to accompany dataset explaining column titles for dataset S3 (S4).

**Author Contributions:** Conceptualization, review, and editing were undertaken by all authors; methodology, analysis, data collation, and preparation of the original draft were undertaken by S.E.D. All authors have read and agreed to the published version of the manuscript.

**Funding:** D.S. was supported by funding from Research England.

**Institutional Review Board Statement:** Not applicable.

**Informed Consent Statement:** Not applicable.

**Data Availability Statement:** The data presented in this study are available in the Supplementary Materials listed above.

**Conflicts of Interest:** The authors declare no conflicts of interest.

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
