# Peer review of "Addressing Threats and Ecosystem Intactness to Enable Action for Extinct in the Wild Species"

_diversity, doi:10.3390/d15020268_

Round 1

Reviewer 1 Report

This is a good article, I am enjoy in reading it.  Some suggestions:

1. Please double check the section of "5. Next steps".  Some statements seemed not complete yet or require more interpretation.

2. Please double check the sentences completeness of the "Possible solution" for "Scenario E" in Table 2.

3. would it be possible to assign all 84 EW species to 1 (or 2) suitable scenario based on information provided in Table 1 and Figure 1?  A table will do.

Author Response

Reviewer 1 responses

Thank you for your attention to our manuscript and the constructive comments you have provided.  On behalf of my co-authors, I have responded to your comments as follows:

  1. Please double check the section of "5. Next steps".  Some statements seemed not complete yet or require more interpretation.

Response to point 1: I have completed a review of section 5 and indeed have noted and completed a number of improvements which I hope have increased the clarity of the text.  These edits are itemised below by line number:

411 - amended the last clause of the sentence to read as follows: "the Wyoming Basin shrub steppe, which encompasses 130,000 km2."

415 - added wording: "indicating a moderate proportion of natural vegetation"

417-419 - added punctuation to improve sentence structure: "Even species that have very degraded ecoregions, with low levels of intact habitat, may be subject to translocation"

419 - deleted 'either'

420-423 - added the following text to clarify the key message from this section: "We recommend that opportunities for recovery of EW species to the wild are thoroughly explored in every case, working within the constraints of site-based conservation, rather than using perceived limitations in habitat availability as a barrier to future action species’ survival."

  1. Please double check the sentences completeness of the "Possible solution" for "Scenario E" in Table 2.

Response to point 2: This sentence has been moved into the main text of the article (as per recommendations from another reviewer) and has been completed as follows: "Reassess suitability for release at regular time points appropriate to the species and in response to changes in the status of potential release sites within, and/or outside of, the species’ indigenous range."

  1. would it be possible to assign all 84 EW species to 1 (or 2) suitable scenario based on information provided in Table 1 and Figure 1?  A table will do.

Response to point 3: We agree that the allocation of all EW species to one or more of the scenarios would be a useful exercise but due to data limitations and potential sensitives of groups currently working with EW species, we instead chose a case study approach.  Case studies allowed us to provide some of the context and pragmatic considerations when applying the scenarios in order to allow practitioners to apply the scenarios themselves.  Most of the authors of this paper are members of the IUCN Extinct in the Wild Task Force and as such we will be systematically assessing EW species in the future.  We intend to work with practitioners already 'on the ground' to refine our understanding of how scenarios might be applied in the future.

Reviewer 2 Report

This is an interesting and generally well-written manuscript that addresses the interesting topic of Extinct in the Wild species. This review focuses on the remaining habitat and how that affects the possibility of reintroduction/reestablishment. It summarizes five different scenarios facing EW species and provides some case studies for each scenario. I note a few issues that need to be addressed, and/or will improve the manuscript.

1. How does the authors’ in press manuscript (reference 7) differ from this submission? I have no way to judge since the in press manuscript is not available, but the authors need to clarify this manuscript is distinct in content and analyses.

 2. Table 2 doesn’t really work as a table. Each scenario should either be described as a paragraph, and the case studies could be incorporated here instead of in a separate section.  Alternatively, the amount of text in the table should be condensed considerably.

3. The number of individuals that ex situ populations were derived from is an extremely important consideration for management/reintroduction but is not mentioned. Does that ex situ collection have sufficient genetic/genotypic diversity to even consider reintroduction? Could estimates of ex situ founder population sizes but incorporated into Table 1?

4. There are many places where scientific names are not italicized. There are also many places where scientific names are needed but not given. I assume the copy editors will make sure these mistakes are corrected.

Minor points

5. A summary of table 1 would be of interest. How many plants versus animals? Among animals, invertebrates versus vertebrates. What group stands out (I was surprised to see so many molluscs on this list). Could plants be further subdivided (monocots vs dicots or angiosperms vs gymnosperms)?

6. L52: what is a “statutory mechanism” in this context?

7. References seem missing for some statements. For example, lines 145-149 has no references for the erroneous classifications and reclassifications. Also, better references could be found for some statements (BirdLife, BirdGuides should be replaced with primary literature)

8. Figure 1: Why are invasive species, genes and diseases lumped into one category? These seem like very distinct threats?

Author Response

Reviewer 2 responses

  1. How does the authors’ in press manuscript (reference 7) differ from this submission? I have no way to judge since the in press manuscript is not available, but the authors need to clarify this manuscript is distinct in content and analyses.

Response: The in press (Smith et al) submission is an evaluation of the ex situ population of each of the EW species rather than the threats operating in the indigenous range.  We have made this distinction clearer in the text in the opening paragraph of section 2.

  1. Table 2 doesn’t really work as a table. Each scenario should either be described as a paragraph, and the case studies could be incorporated here instead of in a separate section.  Alternatively, the amount of text in the table should be condensed considerably.

Response: We take on board these comments and acknowledge that the table has become somewhat lengthier than we originally envisaged.  We feel that the utility of a table in being a ‘quick reference’ to our scenarios is still valuable so we have taken the latter suggestion from Reviewer 2 to reduce the text within the table and transfer much of the content of the table into the main body of the text.

  1. The number of individuals that ex situ populations were derived from is an extremely important consideration for management/reintroduction but is not mentioned. Does that ex situ collection have sufficient genetic/genotypic diversity to even consider reintroduction? Could estimates of ex situ founder population sizes but incorporated into Table 1?

Response: As stated in our response to point 1, the ex situ population is the subject of our soon-to-be-published paper (reference number 6; Smith et al).  Therefore, it would not be appropriate to include an estimate of the ex situ population size until those data are published.

  1. There are many places where scientific names are not italicized. There are also many places where scientific names are needed but not given. I assume the copy editors will make sure these mistakes are corrected.

Response: a thorough proof-read has been undertaken and errors have been corrected in several places in the manuscript including the inclusion of scientific names (although we could find no instances of scientific names that are not italicised).  With regards to using scientific names versus common names, we have tended to use common names where they exist as this improves the readability of the manuscript.  The scientific and common names (where available) are provided at first mention of a species, in Table 1, and for the species featured as case studies, in the subheading of the text.

  1. A summary of table 1 would be of interest. How many plants versus animals? Among animals, invertebrates versus vertebrates. What group stands out (I was surprised to see so many molluscs on this list). Could plants be further subdivided (monocots vs dicots or angiosperms vs gymnosperms)?

Response: thank you for this useful suggestion.  We have included a text summary of the table as an additional sub-section now labelled ‘2.1 Extinct in the Wild Species’.  Subsequent sections have been re-numbered as necessary.

  1. L52: what is a “statutory mechanism” in this context?

Response:

We have added text to clarify that statutory mechanisms prioritise in situ conservation actions.

  1. References seem missing for some statements. For example, lines 145-149 has no references for the erroneous classifications and reclassifications. Also, better references could be found for some statements (BirdLife, BirdGuides should be replaced with primary literature)

Response:

We have added a reference to the Smith et al paper which provides a rationale underpinning deviances from the published EW Red List. The reference from BirdLife International is a species factsheet that is an authoritative providing ecological information on the species and of use in this context.  The BirdGuides article is the most recent report we could find on progress on the recovery of Spix’s macaw.  We do not know if any primary literature on this topic so we feel it is better to cite a recent and reputable source than omit the information altogether.

  1. Figure 1: Why are invasive species, genes and diseases lumped into one category? These seem like very distinct threats?

Response: This is one of the formally-recognised threats in the IUCN’s typology of drivers and mechanisms causing declines in biodiversity; we have replicated the working used by the IUCN.  The threat encompasses problems caused by invasive species, but also invasive genes that are implicated in problems such as genetic swamping, and diseases that have been introduced to novel hosts thereby causing issues with naïve hosts that have not evolved to tolerate such pathogens.  We have attempted to make this clearer in the manuscript so that it is not interpreted as something the study authors have generated, but is instead a recognised threat category.

Reviewer 3 Report

Comments on the MS by Sarah E. Darymple al. entitled: Addressing threats and ecosystem intactness to enable action for Extinct in the Wild species

The MS, submitted to Diversity, addresses an important biodiversity conservation issue, namely, management issues of species extinct in the wild (EW) and discussion of possible scenarios for their reintroduction into the wild. I found the MS to be very well written. The MS was categorized as a review paper, although it seems more like an expert opinion to me, since the goal was not really to provide a literature review, but rather to discuss known facts in order to suggest guidelines for EW reintroduction scenarios. In this context, MS is very relevant as it also appears to be an official statement of the IUCN Species Survival Commission Conservation Translocation Specialist Group. So I have more suggestions for the authors than real corrections. Anyway, I think my points below are important to consider when proposing such important guidelines that I think are very important for the conservation of global biodiversity.

The authors focused only on the species-specific aspects by assessing the chances of EW returning to the wild, but they did not relate this to the fact that this is largely an ecosystem restoration approach, which would be the main purpose of reintroducing EW species to the wild. Thus, in the MS I miss the ecosystem perspective of the EW comeback, especially what the consequences of such a successful comeback might be, since reintroduction areas usually located in areas of high conservation value. All of the EW species listed are low trophic level species. Therefore reintroductions need to be considered in terms of the functional response of potential local predators or the response of other species from the same guild as the reintroduced EW species, leading to increased competition or even indirect competitive effects, i.e. apparent competition, with effects similar to those of alien species introductions. This could be particularly important in ecosystems where species became extinct in the wild long ago.

Another point that might help the authors to better assess the intactness or degradation of ecosystems is the use of biodiversity indicators in the areas where the species of EW used to occur. In this context, the proportion of extinct or critically endangered species may be useful in assessing the extent to which the ecosystem is degraded. A high proportion of EX or CR species might provide a more direct indication of the degree of habitat intactness.

Given Table 1, I wonder if one of the steps for the species whose habitat integrity was rated high, e.g., H±, meaning that there are still many suitable habitats for the species, would be to propose further field studies to confirm that the species is truly extinct in the wild. This could be especially true for invertebrates and plants. Please also comment on this in the Next Steps section on line 394.

Also, Table 1 lists only 83 species, while you claim to have included 84 EW species. Please check. Some plant species are also listed by their genus name only. Why? Please check and correct this. I also did not understand the meaning of the † mark in the table. Are these species then extinct or something else?

I have one more comment on Table 2. In the " Possible Solution" column, I saw that the first paragraph is aimed at making suggestions for rearing and the second for reintroduction in the wild. If this is correct, you should make it clearer, as these two points are very important here. Specific comments on Table 2:

1st row, 3rd column (A): You suggested that reintroductions should be conducted are series of trials. I would like to point out that reintroductions are usually conducted in remote and still more or less preserved ecosystems, where inadequately designed experiments without predictive models for the function of the local ecosystem after release could be dangerous and might threaten other species still living in that ecosystem. Please provide at least a brief indication of this.

3rd row, 3rd column (C): I guess you should achieve naturalization of the species going through the whole process of naturalization, otherwise well known from invasive ecology.

4th row, 3rd column (D): The creation of a habitat means, in fact, the creation of a new ecosystem. Here it is imperative to evaluate the impact of creating a new habitat on other native species in the area, especially so as not to increase the problem of extinction.

5th row, 3rd column (E): I guess the last sentence is not complete in version I got.

The paragraph between lines 213-226 is quite general and vague. I think you need to be more specific here. For example, what does biological understanding actually mean? Ecosystem functioning is probably the most important issue here, but not the only one. Especially the sentence between lines 220 and 223 seems very unclear to me, and I did not understand your point.

When you refer to monitoring (e.g., line 230, Fig. 2), you should be more specific about what kind of monitoring would be needed here. For someone doing a programme to reintroduce EW species, it should be very important what kind of monitoring should be developed, just species presence, abundance, reproduction, survival, monitoring of habitat variables. I think you should propose at least a minimum (i.e. bronze standard) with possible further developments (silver and/or gold standard).

Case studies are of course very useful, but still take up a lot of space in MS. I would prefer to expand the guidance section by providing more details that need to be considered in the measures for EW. In this case, one could probably present fewer species as cases. In general, for each scenario, I would expect a summary or take-home message from the case study species discussed. Some comments on this part:

-          Latin names should be in italic

-          Lines 256-258: I would say that ecosystem approach is necessary here.

-          Line 312: What you meant with mentors? Did you mean indicators?

-          Lines 347-349: Here you are only repeating what you have already said. Please delete this sentence, as it is unnecessary.

-          Lines 356-360: This is highly sensitive since it might have additional ecosystem change consequences. You should at least discuss this here and give your opinion on that (similar in lines 372-373).

-          Lines 377-379: Here you are repeating yourself again. Please, delete this sentence.

In line 395 there is a mistake. Figure 3 instead of 4.

Is Figure 3 really necessary? I could not see any clear point.

Conclusions: I would also note that it would be necessary to establish guidelines to systematically ensure ex situ conservation of CR species. I suspect that not all CR (and EW) are easily reared, and EW are probably only those whose rearing has been possible. This point is very important now to secure ex situ populations for existing CR species. So who should be involved and how should one proceed? Another guiding principle should be that EW reintroduction should occur as soon as possible after extinction in the wild, because post-extinction restructuring of natural communities and coevolutionary processes.

Author Response

Reviewer 3 responses

The MS, submitted to Diversity, addresses an important biodiversity conservation issue, namely, management issues of species extinct in the wild (EW) and discussion of possible scenarios for their reintroduction into the wild. I found the MS to be very well written. The MS was categorized as a review paper, although it seems more like an expert opinion to me, since the goal was not really to provide a literature review, but rather to discuss known facts in order to suggest guidelines for EW reintroduction scenarios. In this context, MS is very relevant as it also appears to be an official statement of the IUCN Species Survival Commission Conservation Translocation Specialist Group. So I have more suggestions for the authors than real corrections. Anyway, I think my points below are important to consider when proposing such important guidelines that I think are very important for the conservation of global biodiversity.

Response: Thank you for the positive comments regarding our manuscript.  If the Editorial team supported a change of article type, as is suggested by the reviewer, we would be happy to take their direction.  According to the ‘instructions to authors’, the article types can either be review articles, or original research articles and there is no ‘opinion’ type of article in this journal.  Consequently, a ‘review’ seemed to be the best choice of article type due to our reliance on synthesising a number of resources to specifically highlight the situation of EW species, rather than generating new data.

The authors focused only on the species-specific aspects by assessing the chances of EW returning to the wild, but they did not relate this to the fact that this is largely an ecosystem restoration approach, which would be the main purpose of reintroducing EW species to the wild. Thus, in the MS I miss the ecosystem perspective of the EW comeback, especially what the consequences of such a successful comeback might be, since reintroduction areas usually located in areas of high conservation value.

Response: Although we are cognisant of the importance of ecosystem restoration as a rationale for reintroductions, in the case of EW species, their recovery in the wild is principally driven by the need to prevent the full extinction of the species, and not as way of achieving the favourable status of an ecosystem.  If ecosystem restoration was the key purpose of a conservation intervention, it might be achieved through more straightforward means than translocating EW species. 

All of the EW species listed are low trophic level species. Therefore reintroductions need to be considered in terms of the functional response of potential local predators or the response of other species from the same guild as the reintroduced EW species, leading to increased competition or even indirect competitive effects, i.e. apparent competition, with effects similar to those of alien species introductions. This could be particularly important in ecosystems where species became extinct in the wild long ago.

Response: We absolutely agree that the reintroduction of EW species would require a full risk assessment of negative interactions with other species in the recipient ecosystem, and that these risks are particularly likely when a species has been extirpated from a system for a considerable amount of time.  However, our intention was not to conduct a full reintroduction/translocation feasibility study but instead flag the avenues available to achieve their recovery.  We have attempted to make this clearer in the latter part of the introduction (section 1) and the opening paragraph of section 2.

Another point that might help the authors to better assess the intactness or degradation of ecosystems is the use of biodiversity indicators in the areas where the species of EW used to occur. In this context, the proportion of extinct or critically endangered species may be useful in assessing the extent to which the ecosystem is degraded. A high proportion of EX or CR species might provide a more direct indication of the degree of habitat intactness.

Response: Thank you for the suggestion.  Although we do not have time available to incorporate these into the current manuscript, as this analysis would constitute major revisions, we will certainly consider biodiversity indicators in our future work on EW species.

Given Table 1, I wonder if one of the steps for the species whose habitat integrity was rated high, e.g., H±, meaning that there are still many suitable habitats for the species, would be to propose further field studies to confirm that the species is truly extinct in the wild. This could be especially true for invertebrates and plants. Please also comment on this in the Next Steps section on line 394.

Response:  This is a salient point and we have added text to the first paragraph of section 5 as suggested.

Also, Table 1 lists only 83 species, while you claim to have included 84 EW species. Please check. Some plant species are also listed by their genus name only. Why? Please check and correct this. I also did not understand the meaning of the † mark in the table. Are these species then extinct or something else?

Response: Thank you for flagging these errors.  They have resulted from a very recent update of the Red List and subsequent last-minute revisions to the manuscript.  As lead author I take full responsibility for missing these mistakes and have corrected them for the revised submission of the manuscript.

I have one more comment on Table 2. In the " Possible Solution" column, I saw that the first paragraph is aimed at making suggestions for rearing and the second for reintroduction in the wild. If this is correct, you should make it clearer, as these two points are very important here.

Response: Reviewers 1 and 2 also flagged issues with this table and I have resolved these and those of reviewer 3 by moving much of the text from column three of the table into the main body of the manuscript.

Specific comments on Table 2:

1st row, 3rd column (A): You suggested that reintroductions should be conducted are series of trials. I would like to point out that reintroductions are usually conducted in remote and still more or less preserved ecosystems, where inadequately designed experiments without predictive models for the function of the local ecosystem after release could be dangerous and might threaten other species still living in that ecosystem. Please provide at least a brief indication of this.

Response: We have added text to section 3.1 to make the requirement for robust trials clearer and also referred to IUCN guidance for translocations and monitoring post-release.

3rd row, 3rd column (C): I guess you should achieve naturalization of the species going through the whole process of naturalization, otherwise well known from invasive ecology.

Response: We are not clear on what the reviewer is requesting here.

4th row, 3rd column (D): The creation of a habitat means, in fact, the creation of a new ecosystem. Here it is imperative to evaluate the impact of creating a new habitat on other native species in the area, especially so as not to increase the problem of extinction.

Response: A reference to risk assessment associated with conservation translocations has been added and the IUCN Guidelines cited.

5th row, 3rd column (E): I guess the last sentence is not complete in version I got.

Response: Apologies, this was an error and has now been rectified.

The paragraph between lines 213-226 is quite general and vague. I think you need to be more specific here. For example, what does biological understanding actually mean? Ecosystem functioning is probably the most important issue here, but not the only one. Especially the sentence between lines 220 and 223 seems very unclear to me, and I did not understand your point.

Response: We have amended the wording to clarify these points raised.  The previous term “biological understanding” has been replaced by “ecological knowledge of the species and recipient ecosystem”.  We have deleted the final sentence of the paragraph.

When you refer to monitoring (e.g., line 230, Fig. 2), you should be more specific about what kind of monitoring would be needed here. For someone doing a programme to reintroduce EW species, it should be very important what kind of monitoring should be developed, just species presence, abundance, reproduction, survival, monitoring of habitat variables. I think you should propose at least a minimum (i.e. bronze standard) with possible further developments (silver and/or gold standard).

Response: We agree with this point and have cited the IUCN Guidelines for Reintroductions and Other Conservation Translocations which has a comprehensive section on monitoring.

Case studies are of course very useful, but still take up a lot of space in MS. I would prefer to expand the guidance section by providing more details that need to be considered in the measures for EW.

Response: We have moved the case studies to follow the first description of each scenario and this has allowed the deletion of text to make each case study more concise. We have also clarified the guidance section as described in comments above.

In this case, one could probably present fewer species as cases. In general, for each scenario, I would expect a summary or take-home message from the case study species discussed.

Response: We did explore the possibility of fewer species and less information per species in earlier versions of our manuscript.  However, when we featured only one species per scenario, it meant that the particular situation of that species obscured the potential for more variation within the scenario – something we felt became a limitation in terms of trying to convey the complexities of working with this challenging set of species.

Some comments on this part:

-          Latin names should be in italic

-          Lines 256-258: I would say that ecosystem approach is necessary here.

-          Line 312: What you meant with mentors? Did you mean indicators?

-          Lines 347-349: Here you are only repeating what you have already said. Please delete this sentence, as it is unnecessary.

-          Lines 356-360: This is highly sensitive since it might have additional ecosystem change consequences. You should at least discuss this here and give your opinion on that (similar in lines 372-373).

-          Lines 377-379: Here you are repeating yourself again. Please, delete this sentence.

Response: Unfortunately, the line numbers do not seem to correspond to the comments and it has been very difficult to ascertain which text needs rectifying.  However, we have corrected all errors where we identified them and the whole manuscript has been through an edit and proof-read to make the writing less repetitive and more concise and accurate.

In line 395 there is a mistake. Figure 3 instead of 4.

Response: now corrected.

Is Figure 3 really necessary? I could not see any clear point.

Response: We felt it useful to include some visualisation of the spatial spread of the species.  With 84 species this became too complicated so we chose to highlight the distribution of those species that had the highest intactness of habitat.  This illustrates the point in the preceding text that highlights the island distributions of some species, and continental range of others.  However, if the editorial team felt that this figure was not necessary in the text, we would be happy to use this figure as a graphical abstract instead.

Conclusions: I would also note that it would be necessary to establish guidelines to systematically ensure ex situ conservation of CR species. I suspect that not all CR (and EW) are easily reared, and EW are probably only those whose rearing has been possible. This point is very important now to secure ex situ populations for existing CR species. So who should be involved and how should one proceed? Another guiding principle should be that EW reintroduction should occur as soon as possible after extinction in the wild, because post-extinction restructuring of natural communities and coevolutionary processes.

Response: We fully agree with these points and have made these in our paper that has recently been accepted in Science (Smith et al 2023).

Round 2

Reviewer 2 Report

The authors have made the appropriate revisions to the manuscript. It is ready for publication in my view.

Reviewer 3 Report

I think the MS is in a good shape and could be accepted. I was not able to judge how many information, if any, is repeating from similar kind of paper authors published recently, Smith et al. 2023, since it is not yet available, but I guess not. Regarding Figure 3, I guess it would be better to use it as a graphical abstract as also suggested by the authors.